# Brachytherapy in the Treatment of Non-Melanoma Skin Peri-Auricular Cancers—A Retrospective Analysis of a Single Institution Experience

**DOI:** 10.3390/cancers14225614

**Published:** 2022-11-15

**Authors:** Mateusz Bilski, Paweł Cisek, Izabela Baranowska, Izabela Kordzińska-Cisek, Nina Komaniecka, Anna Hymos, Ewelina Grywalska, Paulina Niedźwiedzka-Rystwej

**Affiliations:** 1Department of Radiotherapy, Medical University of Lublin, 20-093 Lublin, Poland; 2Department of Brachytherapy, St. John’s Cancer Centre, 20-090 Lublin, Poland; 3Department of Radiotherapy, St. John’s Cancer Centre, 20-090 Lublin, Poland; 4Department of Medical Physics, St. John’s Cancer Centre, 20-090 Lublin, Poland; 5Department of Clinical Oncology, St. John’s Cancer Centre, 20-090 Lublin, Poland; 6Institute of Biology, University of Szczecin, 71-412 Szczecin, Poland; 7Department of Experimental Immunology, Medical University of Lublin, 20-093 Lublin, Poland

**Keywords:** HDR brachytherapy, non-melanoma skin cancer, cancer around the ear, radiation therapy

## Abstract

**Simple Summary:**

Skin cancer is one of the most common cancers worldwide. Non-melanoma skin neoplasms in the head and neck area, the location of lesions around the ear, accounts for approximately 13–15% of all cases. The problematic location of neoplasms within the auricle and around the ear often causes many problems in surgical treatment, which can lead to unsatisfactory cosmetic effects or the presence of positive surgical margins. The presence of positive surgical margins leads to recurrence in about 10–67% of such patients. The aim of the study was to analyse the effectiveness, toxicity profile, and cosmetic effect of two different brachytherapy techniques (contact and interstitial brachytherapy). In our study, we analysed the results of 33 patients treated with HDR contact or interstitial brachytherapy. We showed that this is a highly effective, short, and relatively low burden on patients with cancer of the outer ear, involving the auricle and the skin of the adjacent area. The toxicity of the treatment was low.

**Abstract:**

The location of skin neoplasms in the area of the ears qualifies patients to the so-called high-risk group. The location of neoplasms within the auricle and around the ear often causes many problems in surgical treatment. This is due to the presence of cartilage, the difficulty of performing procedures with obtaining a visually satisfactory cosmetic effect, especially in the presence of extensive lesions and can lead to positive surgical margins which leads to a high risk of recurrence. In such cases, the use of brachytherapy, both as an independent method and as a complementary method after surgery, may be an effective method of local control with an acceptable risk of radiation complications. However, there are no large retrospective studies on the use of brachytherapy in this anatomical region. The aim of the study was to analyse the effectiveness, toxicity profile, and cosmetic effect of two different brachytherapy techniques (contact and interstitial brachytherapy). Methods: This paper presents the results of a retrospective analysis of 33 patients treated with contact or interstitial high-dose-rate (HDR) brachytherapy for skin cancers of the outer ear, involving the auricle and the skin of the adjacent area. Brachytherapy was used both as a definitive treatment (15 patients—43%) and adjuvant treatment after surgery (18 patients—57%). The basic criterion for adjuvant treatment was a positive or narrow (<1 mm) resection margin. Fraction doses from 3 to 7 Gy per fraction were used at intervals from six hours (interstitial brachytherapy) to a maximum of seven days (contact brachytherapy). The treatment time ranged from 1 to 42 days, and the total dose range was 7 to 49 Gy. The follow-up was 29.75 months (range 2–64). Results: In the group of patients treated with adjuvant therapy, in the patients with post-radiation reaction, the mean time from surgery to the start of brachytherapy was 7.72 ± 3.05 weeks, the median was 8 (6–12) weeks, and in the group without post-radiation reaction, the mean time was 11.13 ± 4.41 weeks, the median time was 11 weeks (8–14). The risk of a post-radiation reaction increased significantly more often in patients with more advanced disease. In the case of contact brachytherapy, the post-radiation reaction occurred significantly more often (14/21 patients—43%) than in the case of interstitial brachytherapy (3/11 patients—9.4%). In patients with post-radiation reactions, a significantly larger volume of the skin receiving a dose of 200% was found, and the volume receiving a dose of 150% was close to statistical significance. The mean volume of the skin receiving a 200% dose in the group with post-radiation reactions was 28.05 ± 16.56 cm^3^, the median was 24.86 (0.5–52.3) cm^3^, and the mean volume in the group without post-radiation reaction was 17.98 ± 10.96 cm^3^, median 14.95 (3.9–44.96) cm^3^. The result was statistically significant (Z = 2.035, *p* = 0.041). Conclusion: Interstitial HDR (high-dose-rate) brachytherapy for non-melanoma skin cancers around the ear is highly effective, short, and has a relatively low burden on the patient. The toxicity of the treatment was low. In the case of contact brachytherapy, the toxicity profile is slightly higher but acceptable for patients. This method is preferred in patients in whom interstitial brachytherapy is impossible to perform due to anatomical and logistical reasons. The unquestionable advantage of contact brachytherapy is its ability to be performed on an outpatient basis without the need to stay in the hospital. No severe and late CTCAE ≥III and late RTOG ≥III toxicity was observed. In patients after surgery, in order to minimise the risk of radiation reaction, it is optimal to start treatment at least eight weeks after surgery. In the presence of extensive lesions, the use of interstitial brachytherapy seems to be more advantageous, especially when the expected volume of healthy skin in the dose range of 200% and 150% is above 15 cm^3^ and 50 cm^3^, respectively.

## 1. Introduction

Skin cancer is one of the most common cancers worldwide [1]. The term non-melanoma skin cancers includes the diagnosis of basal cell carcinoma (BCC) and squamous cell carcinoma (SCC) in nearly 99% of cases. Data on the prevalence of its non-melanoma forms are underestimated, but the World Cancer Research Fund reports that in 2020, nearly 1.2 million new cases were diagnosed, of which, over 722,000 were diagnosed in men and over 475,000 were diagnosed in women [2]. These neoplasms are most frequently diagnosed in Australia and the United States of America, and the most diagnoses per 100,000 inhabitants are in Australia and New Zealand, respectively, 140 and 127.5 [1]. Skin cancer in the US is the most common cancer with nearly 3.5 million newly diagnosed patients each year [3]. The latest survey conducted by the European Academy of Dermatology and Venereology (EADV) shows that approximately 7,304,000 Europeans are diagnosed with skin cancer, which is 1.71% of the adult European population [4]. A special group is patients with the location of neoplasms in the area of the ears as well as the central part of the face, which qualifies them for the so-called high-risk groups [5]. In non-melanoma skin neoplasms in the head and neck area, the location of lesions around the ear accounts for approximately 13–15% of all cases [6,7,8]. The problematic location of neoplasms within the auricle and around the ear often causes many problems in surgical treatment. This is due to the presence of cartilage, the difficulty of performing procedures with obtaining a visually satisfactory cosmetic effect, especially in the presence of extensive lesions, and an increased risk of the presence of positive surgical margins. This can lead to relapses and potential wound-healing problems. In patients with localisation of non-melanoma skin cancers within the head and neck, the percentage of incomplete resections ranges from 3–10.8%, depending on the surgical technique [6,7]. According to various sources, the presence of positive surgical margins leads to recurrence in about 10–67% of such patients, compared to 5–14% when the resection is complete [9,10,11]. The use of brachytherapy in this localisation as a method of conservative treatment seems to be a particularly attractive method. Data on the use of brachytherapy in this area are limited and refer to individual cases. Few reports from recent years also include the use of teleradiotherapy in this area [12,13,14,15,16,17]. The potential use of interstitial brachytherapy and/or contact brachytherapy in this group of patients provides issues for comparative analysis and facilitates the selection of the optimal technique. The aim of the study was to analyse the effectiveness, toxicity profile, and cosmetic effect of two different brachytherapy techniques (contact and interstitial brachytherapy).

## 2. Materials and Methods

### 2.1. Patient Characteristics

A group of 33 patients (14 men and 19 women) with skin cancer involving the ear and peri-auricular localisation including the skin of the adjacent area was included in the study (Figure 1). These patients were treated with high-dose-rate (HDR) brachytherapy at the Brachytherapy Department of the Centre of Oncology of the Lublin Region from 2010–2020 (Lublin, Poland). Histopathological confirmation of skin cancer was obtained in all patients. Squamous cell carcinoma dominated (20 patients—57%), followed by basal cell carcinoma (nine patients—26%), and further undifferentiated (four patients—11%). Brachytherapy was used both as a definitive treatment (15 patients—43%) and adjuvant treatment after surgery (18 patients—57%). The basic criterion for adjuvant treatment was a positive or narrow (<1 mm) resection margin. Eligibility for definitive brachytherapy resulted from the patient’s lack of consent to resection (eight patients), the inoperability of the neoplasm (three patients), or a health condition that prevented surgery (four patients). At the time of this analysis, no systemic treatment with checkpoint inhibitors and the Hedgehog pathway was available in Poland.

The tumour stage was determined on the TNM scale of the seventh edition. Lymph node and distant metastases were excluded in all patients. The majority of patients were T2 (15 patients—43%), then T3 (11 patients—31%), and then T1 (seven patients—20%). T4 patients were not eligible for brachytherapy. In the analysed group of patients, contact brachytherapy was dominant (21 patients—66%), and interstitial brachytherapy was used less frequently (11 patients—33%). Patient characteristics are presented in Table 1.

### 2.2. Procedure, Planning, and Treatment

Contact brachytherapy was performed by making an individual applicator from a thermoplastic mask and elastic applicators (by Varian Medical System) sewn or glued on it. The applicators were placed parallel to each other at 0.7–1.5 cm intervals covering the area of the neoplastic infiltration or postoperative scar with a 2 cm margin. The interstitial brachytherapy procedure was performed under local infiltration anaesthesia with the use of 1% Lignocaine. Flexible applicators of 35 cm in length from the Varian Medical System were used. They were inserted 2–5 mm under the skin, parallel to each other according to the Paris system assumptions so that they encompassed the tumour or tumour bed with a healthy skin margin of 1–2 cm. The applicators were placed parallel to each other at a distance of 0.7–1.5 cm. After the contact applicator or the interstitial procedure was performed, a treatment planning tomography was performed with the use of a 32-row Siemens computer tomograph. The layer thickness was from 1–3 mm. Based on the position of the applicators, the CTV (clinical target volume) area was drawn on the tomography scans for treatment planning. In the case of adjuvant brachytherapy, the dose was specified in accordance with the recommendations of GEC ESTRO/ACROP—5 mm from the applicator [18]. In case the thickness of the auricle was less than 5 mm, the dose was applied to its entire thickness. If possible, the applicators were placed on both sides of the auricle. This has allowed to avoid overdose and ensured a homogeneity of the dose. The critical organs were skin, bone, and brain.

### 2.3. Treatment Planning and Dosimetry Analysis

Patients were treated according to various fractionation schedules (Table 2). Fraction doses from 3 to 7 Gy per fraction were used at intervals from six hours (interstitial brachytherapy) to a maximum of seven days (contact brachytherapy). The treatment time ranged from 1 to 42 days, and the total dose range was 7 to 49 Gy. All doses were converted according to the linear-quadratic model into the biologically effective dose (BED) where the alpha/beta ratio was 10 Gy for the tumour and early radiation reaction and 3 Gy for the late radiation reaction. The dosimetric analysis is presented in Table 3. The choice of the scheme was dictated by a number of factors. The shorter schedule was used in elderly patients, in worse general conditions, and in less advanced disease. The shorter interval between fractions concerned interstitial procedure; the longer concerned contact procedure. Contact brachytherapy was selected in lesions with a maximum depth of 3 mm. Interstitial brachytherapy was chosen in lesions with a depth of more than 3 mm. Contact brachytherapy was used preferentially in the case of the auricle. Treatment planning was carried out using the Brachyvision treatment planning system. An Ir 192 source of 0.6 mm diameter and an average activity of 10 Ci was used. Treatment was performed with the 24-channel Gammamed or Gammamed Plus (Varian) apparatus. Figure 2 shows examples of CTV volume, critical organ contouring, and dose distribution with the histogram of one of the treated patients.

### 2.4. Follow-Up after Treatment

Patients in the post-treatment period underwent cyclical clinical evaluation, initially monthly for up to three months; some evaluations also included ENT checking. In selected patients, periodic ultrasounds of the neck and abdominal cavity and chest X-rays were also performed. The patients were assessed for local recurrence, lymph node metastasis, and distant metastasis. The frequency and degree of the severity of early and late radiation reactions were also assessed. Due to the number of patients and the retrospective nature of the data, only the acute reaction was reported. The stage I reaction concerned all patients. In 16 patients (48%), stage II and III reactions were found. No stage IV radiation complications were found in the analysed group of patients. The RTOG scale was used to assess toxicity. Figure 3 illustrates the therapeutic process and the achieved effect during the first six months of follow-up after the completion of brachytherapy treatment in one of the patients.

### 2.5. Statistical Analysis

Depending on the occurrence of a grade ≥II reaction, a statistical analysis of selected parameters for the risk of reaction was performed. When examining the relationship between the variables, the non-parametric Mann–Whitney U test (for independent variables, to compare the differences between the two groups of patients) was used. In order to compare the frequency of the analysed categories depending on the parameters tested, a non-parametric chi-square test was used. The significance level in all tests was *p* = 0.05. Statistical analysis was performed in the Statistica ver. 13.1 and 13.3 (StatSoft Poland).

## 3. Results

In the mean follow-up period of 29.75 months (range 2–64), in one patient (3%), there was a relapse of the disease. It was a local recurrence in the irradiated area. The treatment tolerance was good; only 16 (48%) patients had a clinically significant reaction (grade II and above).

The analysis of the influence of the tested parameters on the occurrence of the post-radiation reaction showed a statistically significant effect of the time from the surgical treatment to the start of irradiation. In the group of patients treated with adjuvant therapy, for the patients with post-radiation reactions, the mean time from surgery to the start of brachytherapy was 7.72 ± 3.05 weeks, and the median was 8 (6–12) weeks, and for the group without post-radiation reactions, the mean time was 11.13 ± 4.41 weeks, and the median time was 11 weeks (8–14). The result was statistically significant (Z = 3.011, *p* = 0.002) (Figure 4). The severity of the post-radiation reaction was also influenced by the tumour stage on the TNM scale (T parameter). The risk of a post-radiation reaction increased significantly more often in patients with more advanced disease (Z = −2.447, *p* = 0.014). The type of brachytherapy used also had an influence on the occurrence of the reaction. In the case of contact brachytherapy, the post-radiation reaction occurred significantly more often (14/21 patients—43%) than in the case of interstitial brachytherapy (3/11 patients—9.4%) Z = 4.50, *p* = 0.339. In the case of contact brachytherapy, the maximum depth of the treated lesion was 5 mm, and the average depth was 3.5 mm. The maximum isodose to the skin was 467.75% (in one pixel), the average isodose was 256%, and the median dose did not exceed 200% isodose.

In patients with post-radiation reaction, a significantly larger volume of the skin receiving the dose of 200% was found, and the volume receiving the dose of 150% was close to statistical significance. The mean volume of the skin receiving the 150% dose in the group with post-radiation reactions was 57.8 ± 13.17 cm^3^, and the median was 59.53 (25.3–77.65) cm^3^, and in the no-post-radiation reaction group, the mean volume was 49.64 ± 12.59 cm^3^, and the median volume was 52.35 (30.1–71.69) cm^3^ (Z = 1.297, *p* = 0.053). The mean volume of the skin receiving the 200% dose in a group with post-radiation reactions was 28.05 ± 16.56 cm^3^, and the median was 24.86 (0.5–52.3) cm^3^, and the mean volume in the group without post-radiation reactions was 17.98 ± 10.96 cm^3^, and the median was 14.95 (3.9–44.96) cm^3^. The result was statistically significant (Z = 2.035, *p* = 0.041) (Figure 5 and Figure 6). The individual results and their statistical significance are presented in Table 4 and Table 5.

### Toxicity

The early and late toxicity of radiotherapy and the application procedure were analysed. Toxicity data were collected for all patients. CTCAE and RTOG scales were used. CTCAE (Common Terminology Criteria for Adverse Events) is an international standard for defining and categorizing adverse events, while RTOG (Toxicity criteria of the Radiation Therapy Oncology Group) is a scale that determines the degree of skin damage after radiotherapy.

Among the 33 analysed patients, 17 (52%) people according to CTCAE and 15 (45.5%) according to RTOG did not present any radiation reaction, and their skin remained unchanged. A total of 14 subjects (42%) had CTCAE grade I, and grade I RTOG was observed in 15 (45.5%) patients. Moderate adverse events occurred in 2 patients (6%) according to CTCAE. RTOG grade II reactions occurred in 3 patients (9%). None of the patients presented a reaction of grade III or higher in both the CTCAE and RTOG scores. A summary of the data on radiation reactions after the applied brachytherapy is presented in Table 6 and Table 7.

## 4. Discussion

Patients with neoplasms of the head and neck region are at risk of late radiation complications, such as tissue fibrosis, oedema, cartilage and/or bone necrosis, and dry mucosa. The risk of radiation complications depends on the irradiated region, the total dose, and the duration of treatment [19]. Radiation complications that arise after radiotherapy of the outer ear occur after the administration of a conventionally fractionated total dose above 60 Gy [20]. These complications include ear pain, tinnitus, hearing impairment, and otitis exudate [21]. There are no new data on radiation complications from the external ear; the old data indicate a risk of cartilage necrosis, especially with lesions greater than 4 cm [22].

In the case of brachytherapy, due to the method of dose distribution, the real range of radiation is small, and it is not possible to exceed the tolerance dose for the middle and inner ear. The heterogeneity of the dose in the area of the auricle still remains a problem as it increases the risk of hot spots in the vicinity of the applicators. For this reason, the use of most of the available applicators (Valencia, Leipzig) or skin electronic brachytherapy does not allow to achieve a good dose distribution [23,24,25,26]. The solution to this problem is the use of individual applicators. In the work of Kuncman et al., individual mould applicators were used, composed of a mixture of paraffin and wax, obtaining a homogeneous dose distribution similar to IMRT [27].

Another solution is to use individual applicators attached to a thermoplastic mask [28]. The application of this method provides, with small and irregular changes, a more conformal dose distribution than electron brachytherapy with high treatment efficacy [29]. In the case of deeper lesions with a very irregular surface and different depths of infiltration, interstitial brachytherapy is used, using plastic tubes in one or more areas [30]. In the analysed group, patients with different severity of acute radiation reactions in a particularly difficult location—the outer ear and adjacent tissues—were compared. A group with a mild reaction and a group with an intense acute radiation reaction were separated and analysed. Factors that influence the presence of a reaction are the time from surgery to irradiation and the volume of high-dose areas, i.e., 150 and 200% of the dose. The technique of brachytherapy also turned out to be important in patients undergoing interstitial brachytherapy; the radiation reaction is lower.

The study also showed that the treatment was well tolerated; it was also characterised by low toxicity. There are no data in the literature describing in such a detailed manner brachytherapy in the area of the outer ear. In the study by Kuncman et al. [27], 10 patients with tumours of the auricle and external nose were irradiated. Dosimetry analysis showed good dose homogeneity in this area, comparable to EBRTF, which may indicate the low toxicity of this treatment. Similarly, in the study by Brovchuk et al., using various brachytherapy techniques, 751 patients were irradiated in the facial area, of which, 36 patients were affected by skin cancer in the area of the outer ear [26]. The treatment tolerance in the whole group was good, acute toxicity of grade II and higher concerned only 48% of patients, and cosmetic effects assessed as good and very good concerned 98% of patients. Contrary to the group of patients analysed in our study, the greater severity of the radiation reaction was observed in patients with squamous cell carcinoma compared to patients with basal cell carcinoma. Additionally, in contrast to the authors’ analysis, the study of Brovchuk et al. did not show any differences in clinical outcomes depending on the applied brachytherapy technique (interstitial, contact) [26].

Similar studies with the use of brachytherapy in skin cancers also indicate the low toxicity of treatment and a good cosmetic effect, although they cover different locations within the head and neck [24,26,30]. The only study on external ear neoplasms deals with LDR (low-dose-rate) brachytherapy and does not include dosimetry analysis. Nevertheless, the results of this study indicate excellent local control and a good cosmetic effect, especially in the case of tumours smaller than 4 cm [22].

## 5. Conclusions

Interstitial HDR (high-dose-rate) brachytherapy for non-melanoma skin cancers around the ear is highly effective and short and has a relatively low burden on the patient. The toxicity of the treatment was low. In the case of contact brachytherapy, the toxicity profile is slightly less favourable but acceptable for patients. This method is preferred in patients in whom interstitial brachytherapy is impossible to perform due to anatomical and logistical reasons. The unquestionable advantage of contact brachytherapy is its ability to be performed on an outpatient basis without the need to stay in the hospital. No severe and late CTCAE ≥ III and late RTOG ≥ III toxicity was observed. In patients after surgery, in order to minimise the risk of radiation reaction, it is optimal to start treatment at least eight weeks after surgery. In the presence of extensive lesions, the use of interstitial brachytherapy seems to be more advantageous, especially when the expected volume of healthy skin in the dose range of 200% and 150% is above 15 cm^3^ and 50 cm^3^, respectively.

## Figures and Tables

**Figure 1 cancers-14-05614-f001:**
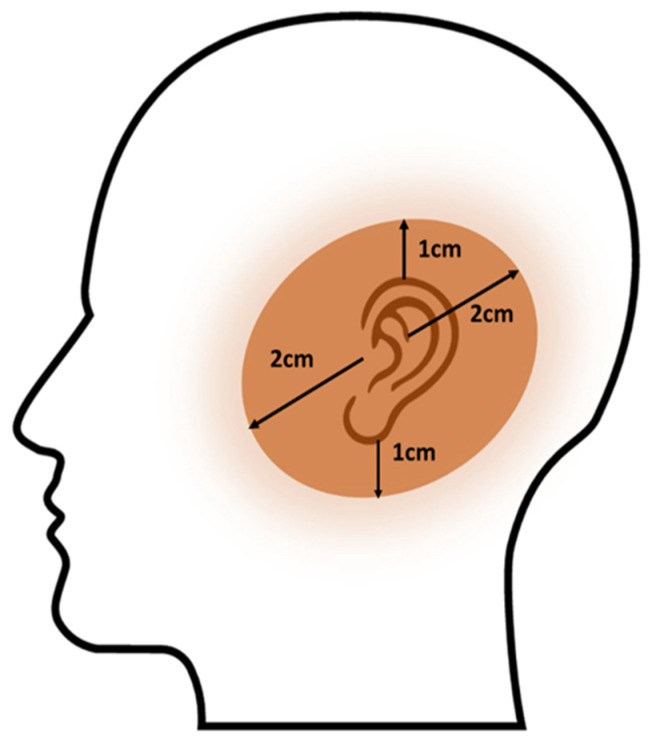
Lesion localisation area in all patients included in the study.

**Figure 2 cancers-14-05614-f002:**
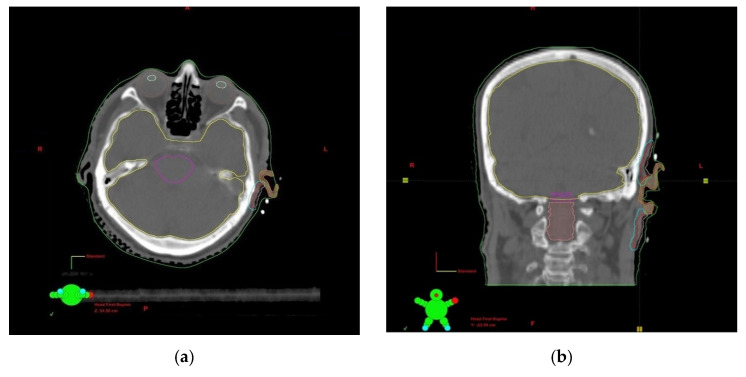
(**a**) Delineation of the brachytherapy target volumes and organs at risk for a patient with non-melanoma skin cancer around the ear, transverse plane; (**b**) Delineation of the brachytherapy target volumes and organs at risk for a patient with non-melanoma skin cancers around the ear, coronal plane; (**c**) The 2D and 3D dose distribution of brachytherapy application; (**d**) A 3D reconstruction of the clinical target volume and CTV surface applicators; (**e**) Dose-volume histogram for target volumes and OARs.

**Figure 3 cancers-14-05614-f003:**
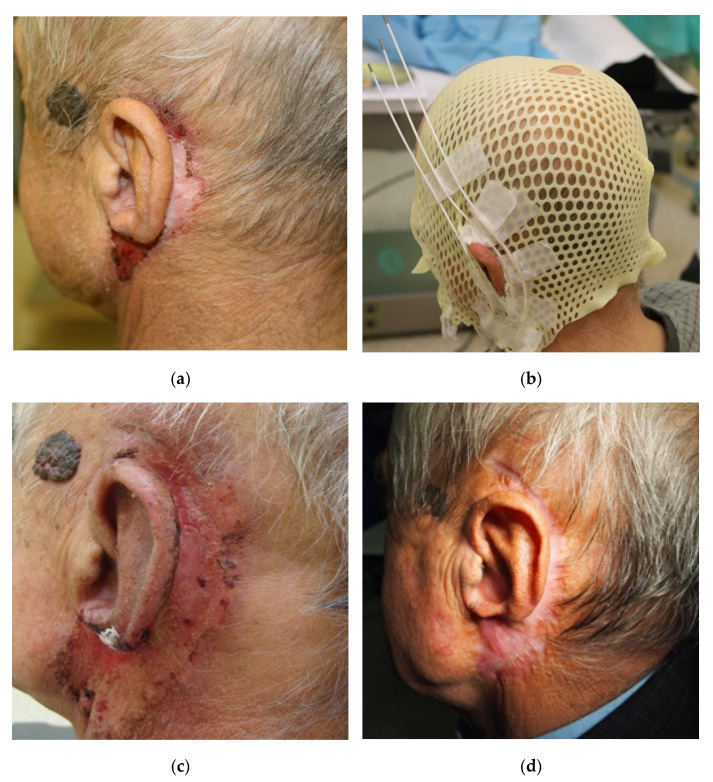
Sample photos of one of the treated patients (the same patient whose plan is presented in Figure 2). (**a**) the image of the primary lesion before the start of treatment; (**b**) the patient with a thermoplastic mask and sewn/glued applicators in the area of the neoplastic lesion; (**c**) the image of an acute radiation reaction, four weeks after the end of brachytherapy; (**d**) the effect obtained six months after the end of brachytherapy.

**Figure 4 cancers-14-05614-f004:**
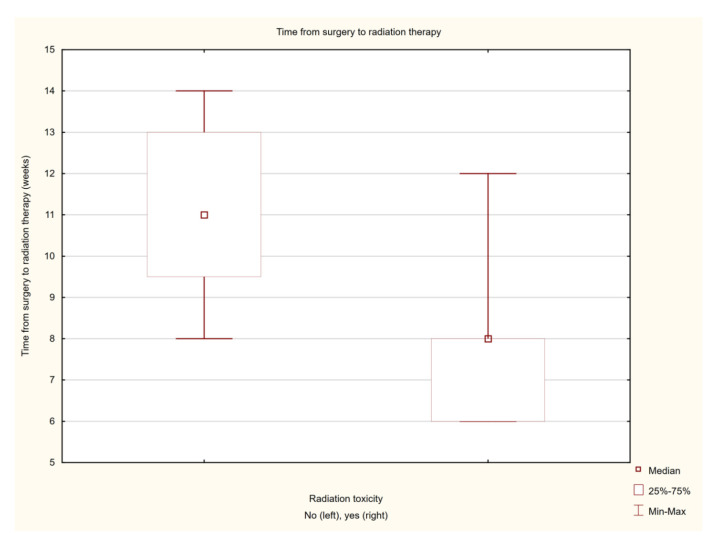
Analysis of the influence of time from surgery on the severity of post-radiation reaction.

**Figure 5 cancers-14-05614-f005:**
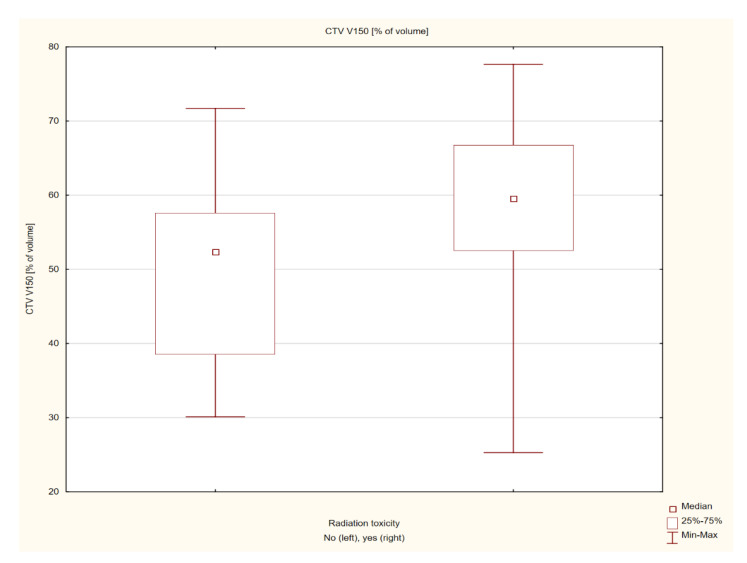
Analysis of the influence of the mean skin volume receiving a dose of 150% on the severity of post-radiation reaction.

**Figure 6 cancers-14-05614-f006:**
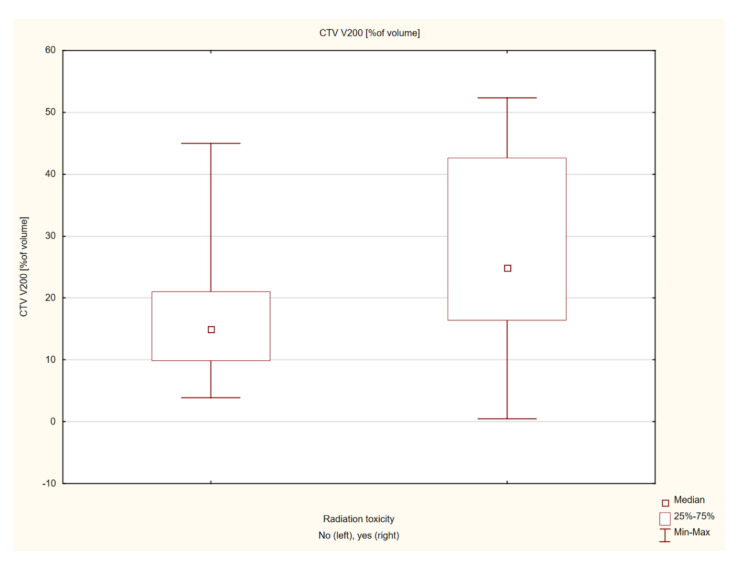
Analysis of the influence of the mean skin volume receiving the 200% dose on the severity of post-radiation reaction.

**Table 1 cancers-14-05614-t001:** Clinical and histopathological characteristics of patients undergoing brachytherapy.

Clinical and Histopathological Factor	Number of Patients (Percentage)	Mean Value	Median Value (Range)
Age (years)	33	80.79 +/− 9.76	83 (63–98)
Gender			
Men	14 (42%)	-	-
Women	19 (58%)	-	-
Histopathological type			
Squamous cell carcinoma	20 (57%)	-	-
Basal cell carcinoma	9 (26%)	-	-
Undifferentiated carcinoma	4 (11%)	-	-
Stage			
T1N0M0	7 (20%)	-	-
T2N0M0	15 (43%)	-	-
T3N0M0	11 (31%)	-	-
Largest lesion size (cm)	-	3.46 +/− 1.79	3 (0–7.5)
Definitive brachytherapy	15 (43%)	-	-
Brachytherapy after surgery	18 (57%)	-	-
Application of brachytherapy			
Contact brachytherapy	21 (66%)	-	-
Interstitial brachytherapy	11 (33%)	-	-
Time from surgery to the start of brachytherapy treatment (weeks)	19	9.16 +/− 2.54	8 (6–14)
Number of applicators	-	6.42 +/− 2.61	6 (3–15)

**Table 2 cancers-14-05614-t002:** Dose fractionation schedules applied in the treatment of all patients.

Dose Fractionation Schedules			
**Total Number of Fractions**	Number of Fractions per Day	Dose per Fraction	Total Dose
15	1	3 Gy	45 Gy
9	2 (interval between fractions of minimum 6 h)	5 Gy	45 Gy
14	2 (interval between fractions of minimum 6 h)	3 Gy	42 Gy
5	1	4 Gy	20 Gy
3	1 every 7 days	7 Gy	21 Gy

**Table 3 cancers-14-05614-t003:** Dosimetry characteristics of doses in particular target volumes and in the brain and associated bone. CTV—clinical target volume, V—volume, D—dose, BED—biologically effective dose.

	Variable	Mean Value	Median Value (Range)
General prescription	Total dose (Gy)	37.73 ± 12.31	45 (7–49)
	Fraction dose (Gy)	3.65 ± 0.88	3.5 (3–7)
	Number of fractions	11.18 ± 4.63	14 (1–15)
	Number of irradiation days	16.15 ± 10.4	19 (1–42)
Target dose	CTV D0.10cc [Gy]	18.14 ± 18.44	12.39 (5.78–88.01)
	CTV D0.10cc [Gy] × number of fractions	164.02 ± 113.61	147 (28.9–645.21)
	CTV D90 [Gy]	4.09 ± 0.87	3.8 (3.1–6.2)
	CTV D90 [Gy] × number of fractions	43.67 ± 17.34	49.5 (6.2–92.55)
	CTV D90 [Gy] × number of fractions BED 10	61.42 ± 25.96	68.78 (10.04–149.65)
	CTV D100[Gy]	2.75 ± 0.56	2.58 (2.04–4.44)
	CTV D100 [Gy] × number of fractions	29.72 ± 12.5	34.35 (3.74–66.6)
	CTV D100 [Gy] × number of fractions	37.93 ± 16.92	42.22 (5.14–96.17)
	CTV V100 [% of volume]	96.76 ± 2.72	97.71 (90.05–99.8)
	CTV V150 [% of volume]	53.6 ± 13.33	53.9 (25.3 −77.65)
	CTV V200 [%of volume]	22.86 ± 14.66	19.49 (0.5–52.31)
OAR’s dose	Brain D1.00cc [Gy]	1.53 ± 0.69	1.49 (0.3–2.94)
	Brain D1.00cc [Gy] × number of fractions	16.46 ± 9.65	14.40 (2.7–33.6)
	Brain D1.00cc [Gy] × number of fractions BED 3	25.95 ± 17.21	21.77 (2.97–58.66)
	Brain D1.00cc [Gy] × number of fractions BED 10	19.31 ± 11.88	17.76 (2.78–41.12)
	Brain D0.10cc [Gy]	1.79 ± 0.83	1.72 (0.34–4.07)
	Brain D0.10cc [Gy] × number of fractions	19.04 ± 10.86	17.25 (3.06–36.9)
	Brain D0.10cc [Gy] × number of fractions BED 3	31.62 ± 20.42	27.77 (3.41–67.13)
	Brain D0.10cc [Gy] × number of fractions BED 10	22.81 ± 13.68	20.54 (3.16–45.98
	Bone D1.00cc [Gy]	2.92 ± 0.93	2.95 (0.82–5.44)
	Bone D1.00cc [Gy] × number of fractions	31.42 ± 14.49	37.20 (4.73–51.6)
	Bone D1.00cc [Gy] × number of fractions BED 3	63.25 ± 32.89	67.92 (9.4–137.65)
	Bone D1.00cc [Gy] × number of fractions BED 10	40.97 ± 19.83	46.42 (6.97–75.59)
	Bone D0.10cc [Gy]	3.63 ± 1.37	3.51 (1.05–8.14)
	Bone D0.10cc [Gy] × number of fractions	38.39 ± 17.91	42.6 (8.14–71.4)
	Bone D0.10cc [Gy] × number of fractions BED 3	86.50 ± 47.15	95.51 (12.75–184.57)
	Bone D0.10cc [Gy] × number of fractions BED 10	52.83 ± 26.38	56.01 (10.44–105.38)

**Table 4 cancers-14-05614-t004:** Influence of selected factors on the severity of radiation reaction on the basis of the Mann–Whitney non-parametric U test. CTV—clinical target volume, D—dose, BED—biologically effective dose.

	Factor	Z	*p*
General information	Age	0.66713	0.504689
	Lesion size	−1.43029	0.152634
	Time from surgery to irradiation	3.01115	0.002603
	Number of applicators	−0.62662	0.530910
Prescription dose	Total dose	−0.34345	0.731262
	Fraction dose	−0.64187	0.520960
	Number of fractions	0.36689	0.713702
	Number of irradiation days	0.54918	0.582880
Target dose	CTV D0.10cc [Gy]	−0.88261	0.377447
	CTV D0.10cc [Gy] × number of fractions	−1.45901	0.144564
	CTV D90 [Gy]	−0.14411	0.885412
	CTV D90 [Gy] × number of fractions	0.21617	0.828857
	CTV D90 [Gy] × number of fractions BED 10	−0.14411	0.885412
	CTV D100 [Gy]	0.55848	0.576517
	CTV D100 [Gy] × number of fractions	0.30621	0.759444
	CTV D100 [Gy] × number of fractions BED 10	0.12609	0.899663
	CTV V100 [% of volume]	1.67530	0.093877
	CTV V150 [% of volume]	−1.92733	0.053939
	CTV V200 [%of volume]	−2.03541	0.041811
OAR’s dose	Brain D1.00cc [Gy]	1.49528	0.134841
	Brain D1.00cc [Gy] × number of fractions	1.53106	0.125756
	Brain D1.00cc [Gy] × number of fractions BED 3	1.74721	0.080602
	Brain D1.00cc [Gy] × number of fractions BED 10	1.53106	0.125756
	Brain D0.10cc [Gy]	1.44112	0.149553
	Brain D0.10cc [Gy] × number of fractions	1.49503	0.134907
	Brain D0.10cc [Gy] × number of fractions BED 3	1.60311	0.108912
	Brain D0.10cc [Gy] × number of fractions BED 10	1.49503	0.134907
	Bone D1.00cc [Gy]	0.68459	0.493605
	Bone D1.00cc [Gy] × number of fractions	1.04481	0.296112
	Bone D1.00cc [Gy] × number of fractions BED 3	1.04481	0.296112
	Bone D1.00cc [Gy] × number of fractions BED 10	1.11687	0.264053
	Bone D0.10cc [Gy]	0.1802	0.856996
	Bone D0.10cc [Gy] × number of fractions	0.9368	0.34886
	Bone D0.10cc [Gy] × number of fractions BED 3	0.86474	0.387181
	Bone D0.10cc [Gy] × number of fractions BED 10	0.86474	0.387181

**Table 5 cancers-14-05614-t005:** Influence of selected factors on the severity of radiation reaction on the basis of the chi-square test. TNM—tumour, node, metastasis.

Feature	Z	*p*
Gender	0.75	0.387
Histopathological type	12.833	0.615
TNM	13	0.602
Indications for brachytherapy—individual vs. adjuvant after surgery	2.53	0.120
Application—contact vs. interstitial	4.5	0.034
Location—auricle vs. adjacent area of the auricle	0.76	0.383

**Table 6 cancers-14-05614-t006:** Characteristics of the early toxicity of radiotherapy and the brachytherapy procedure. CTCAE—Common Terminology Criteria for Adverse Events.

CTCAE (Degree of Severity)	Number of Patients (Percentage)
Grade 0 (none)	17 (52%)
Grade 1 (mild)	14 (42%)
Grade 2 (moderate)	2 (6%)
Grade 3 (severe)	-
Grade 4 (life-threatening)	-
Grade 5 (death)	-

**Table 7 cancers-14-05614-t007:** Characteristics of the toxicity of radiotherapy and the brachytherapy procedure. RTOG—toxicity criteria of the Radiation Therapy Oncology Group.

RTOG	Number of Patients (Percentage)
Grade 0	15 (45.5%)
Grade 1	15 (45.5%)
Grade 2	3 (9%)
Grade 3	-
Grade 4	-

## Data Availability

Due to privacy and ethical concerns, the data that support the findings of this study are available on request from the corresponding author, M.B.

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
