# Peer review of "Brachytherapy in the Treatment of Non-Melanoma Skin Peri-Auricular Cancers—A Retrospective Analysis of a Single Institution Experience"

_cancers, 2022, doi:10.3390/cancers14225614_

Round 1
Reviewer 1 Report
The authors present a paper about "Brachytherapy in the treatment of non-melanoma skin cancers around the ear".
The topic is potentially interesting even though very specific.
The introduction is well written and the materials and methods are adequately clear to understand what the authors have done.
I have a few suggestions that I would like the researchers to address as follows:
1) In this group of high-risk patients multidisciplinary approach is of paramount importance: were they all discussed with the surgeons? which surgeons?
2) One patients had in-field recurrence: it would be extrememly interesting to have more information about this case: which was the histology? which was tthe dose delivered (and fractionation)? type of interventional radiotherapy (brachytherapy)?
3) In cases of contact brachytherapy which was the maximum (and average) depth of lesion treated? Which was the maximum (and average) isodose to the skin?
4) Which was the imaging modality used to stage the patients? MRi or CT?
5) which were the exact criteria to chosse interstitial approach over contact: please make this part more explict.
6) what about the nodal status of these patients? Were they all negative?
7) It would be useful to add a table containing all the doses and fractionations used for these patients
8) It is important to define more clearly, both in the introudction section and in the materials and methods, the anatomical area of interest for this paper.
It is my understing that no ears were included but only areas around the ear: could "peri-auricular" described adequately the group? if so consider to rephrase also the title
9) It would be useful to add a table (or a drawing) with all the sites of localization of the patients included
10) Please modify the conlcusion including also a statement about contact brachytherapy which was presented in this manuscript togheter with interstitial
Author Response
Dear Reviewer,
Thank you for the opportunity to correct our paper. We have thoroughly revised the text and implemented the corrections suggested by the Reviewers. Please find the point-by-point answers to the Reviewers concerns:
- In this group of high-risk patients multidisciplinary approach is of paramount importance: were they all discussed with the surgeons? which surgeons?
All of the patients were discussed on a multidisciplinary meeting with oncological surgeons in our oncologic center. It was done before primary intervention and after surgery with histopathological diagnosis before qualification to brachytherapy.
- One patients had in-field recurrence: it would be extremely interesting to have more information about this case: which was the histology? which was the dose delivered (and fractionation)? type of interventional radiotherapy (brachytherapy)?
It was a patient with non-keratinized squamous cell carcinoma located in the area of the outer ear with embrace of the auricle and part of the left ear canal, with a positive surgical margin after surgery. Contact brachytherapy was performed using an individual applicator for the remaining part of the left external ear canal with 4 flexible applicators. 45 Gy dose was administered in 15 fractions (1 per day). Brachytherapy was started 2 months after the surgery.
3) In cases of contact brachytherapy which was the maximum (and average) depth of lesion treated. Which was the maximum (and average) isodose to the skin? Maximum (and average) isodose to the skin was…
In the case of contact brachytherapy maximum depth of the treated lesion was 5 mm and average depth was 3,5 mm. Maximum isodose to the skin was 467.75% (in one pixel), average isodose was 256% and median dose did not exceed 200% isodose.
4) Which was the imaging modality used to stage the patients?
CT was used for staging. Some of the patients had also undergone MRI if radiologist suggested performing it after CT.
5) which were the exact criteria to choose interstitial approach over contact: please make this part more explicit.
Contact brachytherapy was selected in lesions with maximum depth of 3 mm. Interstitial brachytherapy was chosen in lesions with depth of more than 3 mm. Contact brachytherapy was used preferentially in the case of the auricle.
6) what about the nodal status of these patients? Were they all negative?
All of the patients were nodal negative.
7) It would be useful to add a table containing all the doses and fractionations used for these patients
It was added as Table 2.
Table 2. Dose fractionation schedules applied in the treatment of all patients.
|
Dose fractionation schedules |
|||
|
Total number of fractions |
Number of fractions per day |
Dose per fraction |
Total dose |
|
15 |
1 |
3 Gy |
45 Gy |
|
9 |
2 (interval between fractions of minimum 6 hours) |
5 Gy |
45 Gy |
|
14 |
2 (interval between fractions of minimum 6 hours) |
3 Gy |
42 Gy |
|
5 |
1 |
4 Gy |
20 Gy |
|
3 |
1 every 7 days |
7 Gy |
21 Gy |
8) It is important to define more clearly, both in the introduction section and in the materials and methods, the anatomical area of interest for this paper.
It is now defined more clearly.
It is my understanding that no ears were included but only areas around the ear: could "peri-auricular" described adequately the group? if so consider to rephrase also the title
The term “peri-auricular” has been added to the description of the area and in the title also, thank you.
9) It would be useful to add a table (or a drawing) with all the sites of localization of the patients included
One figure (Figure 1) has been added to visualise better the lesion localization area.
10) Please modify the conclusion including also a statement about contact brachytherapy which was presented in this manuscript together with interstitial
In case of contact brachytherapy, the toxicity profile is slightly higher, but acceptable for patients. This method is preferred in patients in whom intra-tissue brachytherapy is impossible to perform due to anatomical and logistical reasons. The unquestionable advantage of contact brachytherapy is its ability to be performed on an outpatient basis without the need to stay in the hospital.
Thank you again, for all the effort, we do hope, that in the current form the paper can be accepted.
On behalf of the Authors,
Paulina Niedźwiedzka-Rystwej
Reviewer 2 Report
General Comments:
1) the techniques used and the results reported are of relevance to the brachytherapy community
2) Abstract, summary and introduction use very repetitive statements and phrases, which could be better structured not to sounds like the reader is reading the same statements 2-3 times.
3) there is a lack of references on the introductions about previous works on HDR bratty to the ears, please complete a. more thorough literature search and include around line 91. Also, mention how your work differs from the ones already published.
4) Methods in abstract (and maybe manuscript title) should make it clear upfront for the reader that this is a retrospective work.
5) Results in abstract: it is very confusing if the meaning of " time to start brachytherapy". Is this post surgical time? But not all patients had surgery. So what does this time to start mean for the patients that did not have surgery?
6) Line 173: sewn applicators - they seem to have been taped, not sewn, is this correct?
7) Table 2 needs some organization/improved presentation to make it easier to read. Put all the info in a single page, space the rows evenly, center the numbers, rewrite +/- in a more compact fashion, create another column on the left to organize the Data into bigger groups, e.g. "general, prescription, target dose, organs at risk dose, etc". Same goes for the other tables. Consider omitting the top horizontal line and adding a differential line (thicker or double) between the first row (description) and the rest of the table.
8) I believe tables 3, 4 and 6 should have periods and not comas - to follow the notation convention in the rest of the text. A consistent number of significant figures would also make it easier to read here.
9) Lines 26, 307, the word favorable is used here and in other arts of the text however the meaning is vague and not very descriptive of what the author is trying to convey.
10) Some of the discussion brings up previous published data but I believe some of this should be cited in the introduction to make the reader aware of other works in this field and what this particular work brings that is novel and relevant.
Author Response
Dear Reviewer,
Thank you for the opportunity to correct our paper. We have thoroughly revised the text and implemented the corrections suggested by the Reviewers. Please find the point-by-point answers to the Reviewer concerns:
Abstract, summary and introduction use very repetitive statements and phrases, which could be better structured not to sounds like the reader is reading the same statements 2-3 times.
Ad 2
In our opinion, the introduction and discussion are not too repetitive. Nevertheless, we modified the abstract and summary. Moreover, we precised the aim of the study.
Ad 3
3) there is a lack of references on the introductions about previous works on HDR bratty to the ears, please complete a. more thorough literature search and include around line 91. Also, mention how your work differs from the ones already published.
There is a noticeable lack of this kind of research articles, two references were added.
Spigariolo CB, Berti E, Brambilla R, Piccinno R. Radiation therapy of non-melanoma skin cancer of the pinna: an Italian 35-year experience. Ital J Dermatol Venerol. 2022, Volume 157, pp. 92-100.
Foley H, Hopley S, Brown E, Bernard A, Foote M. Conformal orbit sparing radiation therapy: a treatment option for advanced skin cancer of the parotid and ear region. J Med Radiat Sci. 2016, Volume 63, pp. 186-194.
Ad 4
Methods in abstract (and maybe manuscript title) should make it clear upfront for the reader that this is a retrospective work.
Methods and title are now more clear, emphasizing that this is a retrospective work.
Ad 5
5) Results in abstract: it is very confusing if the meaning of " time to start brachytherapy". Is this post surgical time? But not all patients had surgery. So what does this time to start mean for the patients that did not have surgery?
In the group of patients treated with adjuvant therapy, in the patients with post-radiation reaction, the mean time from surgery to the start of brachytherapy was 7.72 ± 3.05 weeks, the median was 8 (6-12) weeks, and in the group without post-radiation reaction, the mean time was 11.13 ± 4.41 weeks, the median time was 11 weeks (8-14).
Ad 6
6) Line 173: sewn applicators - they seem to have been taped, not sewn, is this correct?
Initially they were stitched at large intervals, in fear of pinching the applicator. They were then taped, later only stitched at smaller intervals. In general, the basis was sewing, not gluing, although sometimes additionally sticking a patch was used.
It has been changed in the manuscript.
Ad 7 and Ad 8
7) Table 2 needs some organization/improved presentation to make it easier to read. Put all the info in a single page, space the rows evenly, center the numbers, rewrite +/- in a more compact fashion, create another column on the left to organize the Data into bigger groups, e.g. "general, prescription, target dose, organs at risk dose, etc". Same goes for the other tables. Consider omitting the top horizontal line and adding a differential line (thicker or double) between the first row (description) and the rest of the table.
Currently it is Table 3. It has been improved.
8) I believe tables 3, 4 and 6 should have periods and not comas - to follow the notation convention in the rest of the text. A consistent number of significant figures would also make it easier to read here.
All tables now have periods instead of comas.
Table 3. Dosimetry characteristics of doses in particular target volumes and in the brain and associated bone. CTV – clinical target volume, V – volume, D – dose, BED – biologically effective dose, OAR – organ at risk.
|
|
Variable |
Mean value |
Median value (range) |
|
General prescription |
Total dose (Gy) |
37.73 ± 12.31 |
45 (7 - 49) |
|
|
Fraction dose (Gy) |
3.65 ± 0.88 |
3.5 (3 - 7) |
|
|
Number of fractions |
11.18 ± 4.63 |
14 (1 - 15) |
|
|
Number of irradiation days |
16.15 ± 10.4 |
19 (1 - 42) |
|
Target dose |
CTV D 0.10 cc[Gy] |
18.14 ± 18.44 |
12.39 (5.78 - 88.01) |
|
|
CTV D 0.10 cc[Gy] x number of fractions |
164.02 ± 113.61 |
147 (28.9 - 645.21) |
|
|
CTV D90 [Gy] |
4.09 ± 0.87 |
3.8 (3.1 - 6.2) |
|
|
CTV D90 [Gy] x number of fractions |
43.67 ± 17.34 |
49.5 (6.2 - 92.55) |
|
|
CTV D90 [Gy] x number of fractions BED 10 |
61.42 ± 25.96 |
68.78 (10.04 - 149.65) |
|
|
CTV D100[Gy] |
2.75 ± 0.56 |
2.58 (2.04 - 4.44) |
|
|
CTV D100[Gy] x number of fractions |
29.72 ± 12.5 |
34.35 (3.74 - 66.6) |
|
|
CTV D100[Gy] x number of fractions |
37.93 ± 16.92 |
42.22 (5.14 - 96.17) |
|
|
CTV V100 [% of volume] |
96.76 ± 2.72 |
97.71 (90.05 - 99.8) |
|
|
CTV V150 [% of volume] |
53.6 ± 13.33 |
53.9 (25.3 -77.65) |
|
|
CTV V200 [%of volume] |
22.86 ± 14.66 |
19.49 (0.5 - 52.31) |
|
OAR’s dose |
Brain D1.00cc [Gy] |
1.53 ± 0.69 |
1.49 (0.3 - 2.94) |
|
|
Brain D1.00cc [Gy] x number of fractions |
16.46 ± 9.65 |
14.40 (2.7 - 33.6) |
|
|
Brain D1.00cc [Gy] x number of fractions BED 3 |
25.95 ± 17.21 |
21.77 (2.97 - 58.66) |
|
|
Brain D1.00cc [Gy] x number of fractions BED 10 |
19.31 ± 11.88 |
17.76 (2.78 - 41.12) |
|
|
Brain D0.10cc [Gy] |
1.79 ± 0.83 |
1.72 (0.34 - 4.07) |
|
|
Brain D0.10cc [Gy] x number of fractions |
19.04 ± 10.86 |
17.25 (3.06 - 36.9) |
|
|
Brain D0.10cc [Gy] x number of fractions BED 3 |
31.62 ± 20.42 |
27.77 (3.41 - 67.13) |
|
|
Brain D0.10cc [Gy] x number of fractions BED 10 |
22.81 ± 13.68 |
20.54 (3.16 - 45.98 |
|
|
Bone D1.00cc [Gy] |
2.92 ± 0.93 |
2.95 (0.82 - 5.44) |
|
|
Bone D1.00cc [Gy] x number of fractions |
31.42 ± 14.49 |
37.20 (4.73 - 51.6) |
|
|
Bone D1.00cc [Gy] x number of fractions BED 3 |
63.25 ± 32.89 |
67.92 (9.4 - 137.65) |
|
|
Bone D1.00cc [Gy] x number of fractions BED 10 |
40.97 ± 19.83 |
46.42 (6.97 - 75.59) |
|
|
Bone D0.10cc [Gy] |
3.63 ± 1.37 |
3.51 (1.05 - 8.14) |
|
|
Bone D0.10cc [Gy] x number of fractions |
38.39 ± 17.91 |
42.6 (8.14 - 71.4) |
|
|
Bone D0.10cc [Gy] x number of fractions BED 3 |
86.50 ± 47.15 |
95.51 (12.75 - 184.57) |
|
|
Bone D0.10cc [Gy] x number of fractions BED 10 |
52.83 ± 26.38 |
56.01 (10.44 - 105.38) |
Table 4. Influence of selected factors on the severity of radiation reaction on the basis of the Mann-Whitney non-parametric U test. CTV – clinical target volume, D – dose, BED – biologically effective dose.
|
|
Factor |
Z |
p |
|
General information |
Age |
0.66713 |
0.504689 |
|
|
Lesion size |
-1.43029 |
0.152634 |
|
|
Time from surgery to irradiation |
3.01115 |
0.002603 |
|
|
Number of applicators |
-0.62662 |
0.530910 |
|
Prescription dose |
Total dose |
-0.34345 |
0.731262 |
|
|
Fraction dose |
-0.64187 |
0.520960 |
|
|
Number of fractions |
0.36689 |
0.713702 |
|
|
Number of irradiation days |
0.54918 |
0.582880 |
|
Target dose |
CTV D 0.10 cc[Gy] |
-0.88261 |
0.377447 |
|
|
CTV D 0.10 cc[Gy] x number of fractions |
-1.45901 |
0.144564 |
|
|
CTV D90 [Gy] |
-0.14411 |
0.885412 |
|
|
CTV D90 [Gy] x number of fractions |
0.21617 |
0.828857 |
|
|
CTV D90 [Gy] x number of fractions BED 10 |
-0.14411 |
0.885412 |
|
|
CTV D100[Gy] |
0.55848 |
0.576517 |
|
|
CTV D100[Gy] x number of fractions |
0.30621 |
0.759444 |
|
|
CTV D100[Gy] x number of fractions BED 10 |
0.12609 |
0.899663 |
|
|
CTV V100 [% of volume] |
1.67530 |
0.093877 |
|
|
CTV V150 [% of volume] |
-1.92733 |
0.053939 |
|
|
CTV V200 [%of volume] |
-2.03541 |
0.041811 |
|
OAR’s dose |
Brain D1.00cc [Gy] |
1.49528 |
0.134841 |
|
|
Brain D1.00cc [Gy] x number of fractions |
1.53106 |
0.125756 |
|
|
Brain D1.00cc [Gy] x number of fractions BED 3 |
1.74721 |
0.080602 |
|
|
Brain D1.00cc [Gy] x number of fractions BED 10 |
1.53106 |
0.125756 |
|
|
Brain D0.10cc [Gy] |
1.44112 |
0.149553 |
|
|
Brain D0.10cc [Gy] x number of fractions |
1.49503 |
0.134907 |
|
|
Brain D0.10cc [Gy] x number of fractions BED 3 |
1.60311 |
0.108912 |
|
|
Brain D0.10cc [Gy] x number of fractions BED 10 |
1.49503 |
0.134907 |
|
|
Bone D1.00cc [Gy] |
0.68459 |
0.493605 |
|
|
Bone D1.00cc [Gy] x number of fractions |
1.04481 |
0.296112 |
|
|
Bone D1.00cc [Gy] x number of fractions BED 3 |
1.04481 |
0.296112 |
|
|
Bone D1.00cc [Gy] x number of fractions BED 10 |
1.11687 |
0.264053 |
|
|
Bone D0.10cc [Gy] |
0.1802 |
0.856996 |
|
|
Bone D0.10cc [Gy] x number of fractions |
0.9368 |
0.34886 |
|
|
Bone D0.10cc [Gy] x number of fractions BED 3 |
0.86474 |
0.387181 |
|
|
Bone D0.10cc [Gy] x number of fractions BED 10 |
0.86474 |
0.387181 |
Table 5. Influence of selected factors on the severity of radiation reaction on the basis of the chi-square test. TNM – tumour, node, metastasis.
|
Feature |
Z |
P |
|
Gender |
0.75 |
0.387 |
|
Histopathological type |
12.833 |
0.615 |
|
TNM |
13 |
0.602 |
|
Indications for brachytherapy – individual vs. adjuvant after surgery |
2.53 |
0.120 |
|
Application – contact vs. interstitial |
4.5 |
0.034 |
|
Location – auricle vs. adjacent area of the auricle |
0.76 |
0.383 |
Table 6. Characteristics of the early toxicity of radiotherapy and the brachytherapy procedure. CTCAE – Common Terminology Criteria for Adverse Events.
|
CTCAE (degree of severity) |
Number of patients (percentage) |
|
Grade 0 (none) |
17 (52%) |
|
Grade 1 (mild) |
14 (42%) |
|
Grade 2 (moderate) |
2 (6%) |
|
Grade 3 (severe) |
- |
|
Grade 4 (life-threatening) |
- |
|
Grade 5 (death) |
- |
Table 7. Characteristics of the toxicity of radiotherapy and the brachytherapy procedure. RTOG – toxicity criteria of the Radiation Therapy Oncology Group.
|
RTOG |
Number of patients (percentage) |
|
Grade 0 |
15 (45.5 %) |
|
Grade 1 |
15 (45.5 %) |
|
Grade 2 |
3 (9 %) |
|
Grade 3 |
- |
|
Grade 4 |
- |
Ad9
9) Lines 26, 307, the word favorable is used here and in other arts of the text however the meaning is vague and not very descriptive of what the author is trying to convey.
We changed sentences containing this word to: The toxicity of the treatment was low.
Ad 10
10) Some of the discussion brings up previous published data but I believe some of this should be cited in the introduction to make the reader aware of other works in this field and what this particular work brings that is novel and relevant.
It is now cited in the introduction.
Thank you again, for all the effort, we do hope, that in the current form the paper can be accepted.
On behalf of the Authors,
Paulina Niedźwiedzka-Rystwej
Reviewer 3 Report
The authors should specify if contact brachytherapy was performed only as post-surgery treatment or also as exclusive treatment. If contact brachytherapy was used as exclusive treatment how much was the margins? Usually, the margins have to be defined on the basis of histology and tumor characteristics. Furthermore, has the dose been prescribed to to 3–5mm under the skin surface? Was the distance from the source to the skin of 5mm is to obtain homogeneity on the surface of the skin, and avoid an overdosage? A total HDR dose equivalent to about 60 Gy LDR (prescribed at 5 mm depth) is recommended. (GEC-ESTRO ACROP recommendations in skin brachytherapy).
The authors may underline as skin brachytherapy should be very helpful in old patients (In Vivo 2021; 35(4):2313-2319;doi: 10.21873/invivo.12505.; Biomed Res Int 2018; 2018:2178469; doi: 10.1155/2018/2178469).
Author Response
Dear Reviewers,
Thank you for the opportunity to correct our paper. We have thoroughly revised the text and implemented the corrections suggested by the Reviewers. Please find the point-by-point answers to your concerns:
The authors should specify if contact brachytherapy was performed only as post-surgery treatment or also as exclusive treatment. If contact brachytherapy was used as exclusive treatment how much was the margins? Usually, the margins have to be defined on the basis of histology and tumor characteristics. Furthermore, has the dose been prescribed to to 3–5mm under the skin surface? Was the distance from the source to the skin of 5mm is to obtain homogeneity on the surface of the skin, and avoid an overdosage? A total HDR dose equivalent to about 60 Gy LDR (prescribed at 5 mm depth) is recommended. (GEC-ESTRO ACROP recommendations in skin brachytherapy).
The authors may underline as skin brachytherapy should be very helpful in old patients (In Vivo 2021; 35(4):2313-2319;doi: 10.21873/invivo.12505.; Biomed Res Int 2018; 2018:2178469; doi: 10.1155/2018/2178469).
Yes, we used the ESTRO/ACROP recommendations, we added one reference:
Guinot JL, Rembielak A, Perez-Calatayud J, Rodríguez-Villalba S, Skowronek J, Tagliaferri L, Guix B, Gonzalez-Perez V, Valentini V, Kovacs G; GEC ESTRO. GEC-ESTRO ACROP recommendations in skin brachytherapy. Radiother Oncol. 2018, Volume 126, pp. 377-385.
We modified the text according to the suggestions.
Thank you again, for all the effort, we do hope, that in the current form the paper can be accepted.
On behalf of the Authors,
Paulina Niedźwiedzka-Rystwej
Round 2
Reviewer 1 Report
The authors have satisfactorily addressed all of my suggestions. I have no further comments.
Author Response
Thank you.